# Dynamics of *Trypanosoma cruzi* infection in hamsters and novel association with progressive motor dysfunction

Harry Langston[1¤a], Amanda Fortes Francisco[1], Ciaran Doidge[1¤b], Chrissy H. Roberts[2], Archie A. Khan[1], Shiromani Jayawardhana[1], Martin C. Taylor[1], John M. Kelly[1], Michael D. Lewis [1¤c] *

1 Department of Infection Biology, London School of Hygiene and Tropical Medicine, London, United Kingdom, 2 Department of Clinical Research, London School of Hygiene and Tropical Medicine, London, United Kingdom

¤a Current address: Imperial College London, London, United Kingdom
¤b Current address: Barts Cancer Institute, Queen Mary University of London, London, United Kingdom
¤c Current address: Division of Biomedical Sciences, Warwick Medical School, University of Warwick, Coventry, United Kingdom
* michael.d.lewis@warwick.ac.uk

**Data Availability Statement:** All relevant data are within the manuscript and its Supporting Information files.

## Abstract

Chagas disease is a zoonosis caused by the protozoan parasite *Trypanosoma cruzi*. Clinical outcomes range from long-term asymptomatic carriage to cardiac, digestive, neurological and composite presentations that can be fatal in both acute and chronic stages of the disease. Studies of *T. cruzi* in animal models, principally mice, have informed our understanding of the biological basis of this variability and its relationship to infection and host response dynamics. Hamsters have higher translational value for many human infectious diseases, but they have not been well developed as models of Chagas disease. We transposed a real-time bioluminescence imaging system for *T. cruzi* infection from mice into female Syrian hamsters (*Mesocricetus auratus*). This enabled us to study chronic tissue pathology in the context of spatiotemporal infection dynamics. Acute infections were widely disseminated, whereas chronic infections were almost entirely restricted to the skin and subcutaneous adipose tissue. Neither cardiac nor digestive tract disease were reproducible features of the model. Skeletal muscle had only sporadic parasitism in the chronic phase, but nevertheless displayed significant inflammation and fibrosis, features also seen in mouse models. Whereas mice had normal locomotion, all chronically infected hamsters developed hindlimb muscle hypertonia and a gait dysfunction resembling spastic diplegia. With further development, this model may therefore prove valuable in studies of peripheral nervous system involvement in Chagas disease.

## Author summary

Chagas disease is a caused by American trypanosomes (*Trypanosoma cruzi*). These are microscopic parasites that circulate in wild mammals across most of the Americas and

**Funding:** The research was supported by UK Medical Research Council (https://www.ukri.org/councils/mrc/) grants MR/T015969/1 to J.M.K. and MR/R021430/1 to M.D.L. No authors received salary directly from the funders. The funders had no role in study design, data collection and analysis, decision to publish, or preparation of the manuscript.

**Competing interests:** The authors have declared that no competing interests exist.

can also be transmitted to humans. Much of our knowledge about how *T. cruzi* causes Chagas disease comes from studies of infections in mice, but the data do not capture the full range of clinical outcomes seen in humans. For many other pathogens the hamster has proved to be a valuable model of human infections. We therefore aimed to apply some of the latest advances in *T. cruzi* infection imaging technology to studies in this alternative experimental model. In the early stages, parasites were widely disseminated throughout the body, but after several months parasites became almost entirely restricted to the skin. Hamsters did not show signs of heart or gut disease, which are common in humans, but they did develop skeletal muscle pathology. Stiffness in the hindlimbs grew progressively worse and resulted in a visibly altered gait, suggestive of damage to the nervous system. With further development, this model may therefore prove valuable in studies of peripheral nervous system involvement in Chagas disease.

## Introduction

Chagas disease (American trypanosomiasis) is caused by infection with *Trypanosoma cruzi*, a protozoan parasite. Approximately 6 million people are infected and the disease causes ~10,000 deaths annually as well as a large morbidity burden in affected populations [1]. Clinical outcomes are highly heterogeneous, encompassing muscle and nervous tissue pathologies affecting the heart and gastro-intestinal tract, as well as long-term asymptomatic carriage [2] Pathogenesis is thought to be primarily a result of collateral damage to infected tissues from the host's cellular immune response, which leads to fibrosis, microvascular abnormalities, denervation and consequent organ dysfunction [3–5].

Animal infection models are critical to research on the biological basis of Chagas disease. *T. cruzi* is a very cosmopolitan parasite in its natural host range [6] and a variety of species have been used in experimental settings, including rats, rabbits, dogs, guinea pigs, opossums and non-human primates, often reproducing important features of the human disease spectrum [7]. Mice, however, are by far the most widely used species; in most cases they develop inflammatory cardiomyopathy [8–14] and a few models are suitable for studying aspects of digestive disease [15–18]. The compatibility of murine models with technologies such as bioluminescence imaging (BLI), which enable real-time monitoring of infections, is a major advantage and has led to significant advances in understanding how spatio-temporal infection dynamics relate to pathology development [9,15,19]. Mice have been less well developed as models of advanced digestive megasyndromes, peripheral neuropathy or CNS infection, an important cause of acute mortality in human patients [20,21]. Furthermore, the use of a few inbred mouse strains limits our ability to study the impact of host genetic diversity on disease pathogenesis and clinical outcome heterogeneity.

The aim of this study was to develop an *in vivo/ex vivo* BLI system for *T. cruzi* infection in the Syrian hamster, *Mesocricetus auratus*. Hamsters can show closer alignment with human diseases than mice for a variety of infectious agents, for example *Clostridium difficile*, SARS-CoV-2, West Nile Virus, adenoviruses, *Leishmania donovani* and *Schistosoma haematobium* [22–26]. Hamsters have also been described as useful models of Chagas cardiac disease [27–33] and in some cases are reported to develop a dilated large intestine indicative of megacolon [28]. They are also one of the few species readily available, besides mice, whose physical size is compatible with standard *in vivo* imaging chambers. We therefore reasoned that a hamster *T. cruzi* BLI system could add value to the landscape of experimental Chagas disease *in vivo* models.

## Methods

### Ethics statement

Animal work was approved by the London School of Hygiene and Tropical Medicine Animal Welfare and Ethical Review Board and carried out under UK Home Office project licence (PPL 70/8207) in accordance with the UK Animals (Scientific Procedures) Act.

### Parasites, animals and infections

*T. cruzi* CL Brener (genetic lineage TcVI) constitutively expressing the red-shifted firefly luciferase gene *PpyRE9h* [34] were used in all experiments. Infectious trypomastigotes were generated as previously described [34]. Animals used were female CB17 SCID mice aged 8–10 weeks (bred in house), female BALB/c, C57BL6 and C3H/HeN mice aged 6–8 weeks (Charles River) and female Syrian hamsters aged 6–8 weeks old (Janvier Labs). Animals were acclimatised to the vivarium for 1–2 weeks prior to use in experiments. SCID mice were infected with $10^4$ trypomastigotes and all other mice were infected with $10^3$ trypomastigotes. Hamsters were infected with 5 x $10^3$ or $10^5$ trypomastigotes. Inocula were given as intra-peritoneal (i.p.) injections of 0.2 mL in sterile 1X Dulbecco′s Phosphate Buffered Saline (DPBS).

Animals were maintained under specific pathogen-free conditions in individually ventilated cages, with 2–3 hamsters and 5–6 mice per cage, with a 12 hour light/dark cycle and *ad libitum* food and water. An investigator who was blinded to the infection/control group allocations assigned individual animals to cages. No randomisation procedure was used to allocate animals to infection or control groups. Animals were checked visually twice daily and weighed on average every two weeks. Humane end-points were any of the following: loss of >20% body weight, unwillingness to move, reluctance to feed and drink freely for >6 hours, loss of balance. No animals in this study reached any of the humane end-points.

### Bioluminescence imaging

Hamsters were injected with 150 mg/kg d-luciferin i.p., then anaesthetized using 2.5% (v/v) gaseous isoflurane in oxygen. To measure bioluminescence, hamsters were placed in a Lumina II In Vivo Imaging System (IVIS) (PerkinElmer) and images were acquired 10–20 minutes after d-luciferin administration using LivingImage v4.7 (PerkinElmer). A custom 3D printed nosecone was used for continuous isoflurane delivery during imaging sessions (S1 Data). Exposure times varied between 30 seconds and 5 minutes, depending on bioluminescence signal intensity. After imaging, animals were revived and returned to cages. For *ex vivo* imaging, hamsters were injected with 150 mg/kg d-luciferin i.p., then sacrificed by ex-sanguination under terminal anaesthesia (400–600 mg/kg Pentobarbital Sodium) 7 minutes later. Trans-cardiac perfusion was performed with 20 mL 0.3 mg/mL d-luciferin in PBS. Organs and tissues were transferred to culture dishes, soaked in 0.3 mg/mL d-luciferin in PBS, and then imaged using the IVIS Lumina II. One hamster from the low inoculum group was excluded from further analysis because a bioluminescence signal was never detected, indicating that infection did not become established.

To estimate total parasite loads in live hamsters, regions of interest (ROIs) were drawn using Living Image v4.7 to quantify bioluminescence expressed as total flux (photons/second [p/s]). The detection threshold for *in vivo* imaging was determined using uninfected control hamsters [34].

*Ex vivo* images of tissues and organs were scored for the presence of *T. cruzi* using a detection threshold for infection foci of at least 10 contiguous bioluminescent pixels of radiance $\geq 3 \times 10^3$ p/s/cm$^2$/sr [19]. Ex vivo parasite loads were inferred from tissue/organ-specific bioluminescence total flux (p/s) after subtracting the mean + 2SDs of matching samples from

uninfected control hamsters (n = 7) to account for differences in background luminescence signal amongst different tissue and organ types. Tissue/organ-specific infection intensities were calculated as the fold change in bioluminescence radiance (p/s/cm$^2$/sr), which normalises for differences in tissue sample surface areas, compared with the mean radiance for the matching tissue/organ type measured for uninfected control hamsters (n = 7).

## Histopathology

Tissue samples were fixed in Glyo-Fixx (Epredia) for 24–72 hours, then dehydrated in an ethanol series, cleared in xylene (Sigma), and embedded in paraffin for histomorphometric analysis [9,19]. Five-micron tissue sections were stained with Haematoxylin and Eosin (H&E), picrosirius red or Masson's trichrome and analysed using a DFC295 camera attached to a DM3000 light-emitting diode microscope (Leica) as described [9,19,35,36]. Images were acquired of 10–15 randomly selected microscope fields of view (FOV, area at 400X magnification = 2.66 x 10$^5$ μm$^2$) per animal. Images were digitised using Leica Application Suite v4.5.0. For the analysis of tissue inflammatory infiltrates, the total number of nuclei were counted in H&E-stained sections. Myositis (skeletal muscle) and myocarditis (heart) scores were calculated as the average cellularity (mean number of nuclei per FOV) for each animal. A significant increase in cellularity compared to uninfected controls was considered indicative of inflammation. For the fibrosis index, the percentage of total FOV area positive for collagen in picrosirius red- or Masson's trichrome stained sections (red or blue pixels respectively) was quantified, also in Leica Application Suite v4.5.0. A significant increase in collagen content compared to uninfected controls was considered indicative of fibrosis.

## Limb adduction evaluation

Images from the *in vivo* imaging sessions were adjusted in Living Image v4.7 (PerkinElmer) to remove the bioluminescence overlay. The line measurement tool was then used to measure the distance between the centres of the two forepaws and between the centres of the two hindpaws.

## Statistics

Individual animals were used as the unit of analysis. Groups were compared using Student's *t*-test or one-way ANOVA, with Tukey's post-hoc correction in GraphPad Prism v.8.

## Results

### *In vivo* bioluminescence imaging permits long-term tracking of *T. cruzi* infection in hamsters

We infected hamsters with either a low (5,000) or high (100,000) inoculum of *T. cruzi* TcVI-CLBR trypomastigotes expressing the red-shifted luciferase reporter *PPy*RE9h [34]. The resulting infections were monitored regularly by *in vivo* bioluminescence imaging (Fig 1A and 1B). For both inocula, parasite burdens proceeded through a typical acute phase wave that transitioned into a stable, long-term chronic infection. In the high inoculum group, the acute phase was 3-fold more intense and its peak occurred at 2 weeks post-infection (p.i.) rather than 4 weeks p.i. for the low inoculum group (Fig 1B). Spatially, *in vivo* imaging indicated that the acute phase infection was highly concentrated near to the injection site in the abdomen, but from 8 weeks p.i. onwards, the bioluminescence signal became widely distributed, multifocal and dynamic between analysis time-points (Fig 1A). The size of the inoculum did not affect the level at which chronic parasite burdens stabilised, nor the spatio-temporal

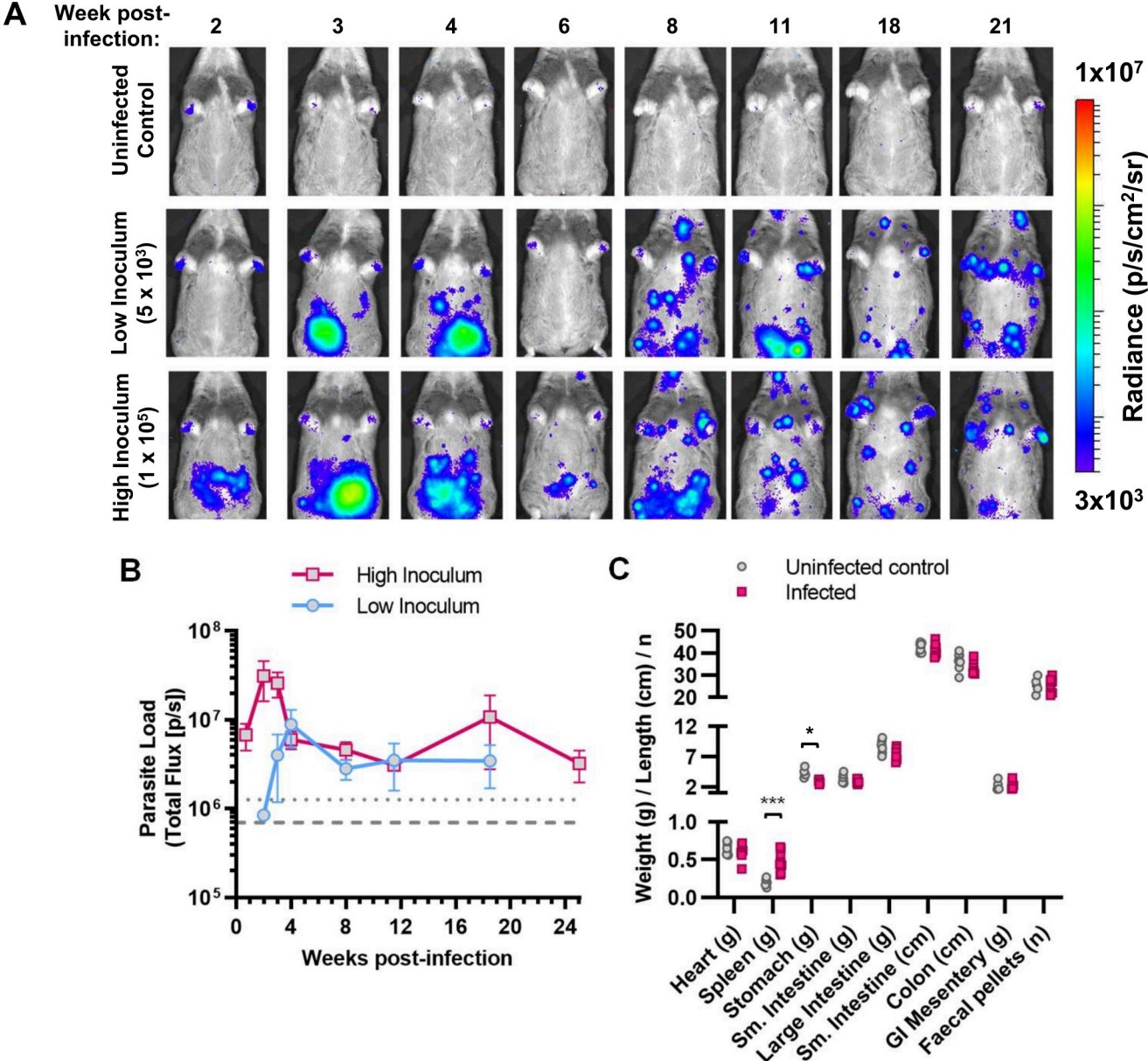

**Fig 1. Serial evaluation of *T. cruzi* CLBR infection in Syrian hamsters by *in vivo* bioluminescence imaging. A, B:** Course of transgenic bioluminescent TcVI-CLBR infection in female Syrian hamsters tracked by *in vivo* imaging. **A:** Panels show an example series of ventral images for one hamster per group taken at the indicated times post-infection. Log-scale pseudocolour heat-map shows intensity of bioluminescence as a proxy for parasite load; minimum and maximum radiances are indicated. **B**: Chart shows the changes in mean ±S.E.M., total bioluminescence over time for hamsters inoculated with 5 x 10$^3$ (low inoculum, n = 3) or 1 x 10$^5$ (high inoculum, n = 8) parasites. Threshold lines are the mean (dashes) and the mean +2SD (dots) of background signal obtained for uninfected control hamsters (n = 7). **C:** Evaluation of organ sizes/weights and the number of faecal pellets present in the colon at 152–182 days post-infection (high inoculum group (n = 8–9 with the exception of GI mesentery, n = 5, and faecal pellets n = 6), uninfected control group (n = 8, with the exception of GI mesentery and faecal pellets, n = 4). Asterisks indicate *p*-values for comparisons with the uninfected control group in multiple unpaired *t*-tests (*** *p*<0.001, * *p*<0.05).

distribution of the infection. For further analyses, we focussed on the high inoculum approach because acute infections in the low inoculum group tended to be more variable.

Monitoring of animals identified gait abnormalities gradually developing during the chronic phase of infection, with pronounced hindlimb rigidity. Muscle mass was observed to

be within normal limits at post-mortem. With respect to potential features of human Chagas disease, we observed significant splenomegaly, but no sign of cardiomegaly, digestive mega-syndromes or constipation at the experimental end-point of 152–182 days p.i. (Fig 1C).

## Skin is the primary reservoir of chronic parasite persistence in hamsters

*T. cruzi* displays a very broad cell and tissue tropism, but *in vivo* imaging does not permit unambiguous localisation of signals to organs. Therefore, to better understand infection dynamics in this model we conducted *ex vivo* bioluminescence imaging post-mortem, at the acute-chronic transition (34–38 days p.i.) and in the established chronic phase (152–182 days p.i.) (Fig 2A). At 34–38 days p.i., infection was almost universally detected across all the tissues and organs that were sampled (Fig 2A, 2B, 2D and 2F). In terms of absolute parasite loads (total bioluminescent flux/sec), the skin harboured the greatest proportion of parasites (58.0% ±19.2%) followed by several fat-rich organs each containing 5–10% of the total (GI mesentery, genito-urinary system, subcutaneous adipose, visceral adipose) (Fig 2D). Infection intensity, which takes into account the large variation in organ size, varied over more than three orders of magnitude. Of note, parasite loads measured in this way were relatively high in the skin, adipose, GI mesentery and genito-urinary system, moderately high in striated muscles (heart, skeletal muscle, abdomen/peritoneum) and relatively low in the GI tract, liver and spleen (Figs 2F and S1).

In the established chronic phase, infection became undetectable in the majority of sites in the majority of individual animals (Fig 2A, 2C, 2E and 2G). It should be noted that the limit of detection is <20 parasites in the mouse colon [37], but the larger size of hamster organs likely reduces detection sensitivity. Persistent infection was mainly restricted to the skin and subcutaneous adipose tissue, with sporadic, usually small foci of infection in a few other organs, most often the skeletal muscle, spleen and lungs. In contrast to mouse models [9,15], parasites were only rarely detected in the GI tract during the chronic stage. Considering absolute numbers, the skin harboured a large majority of the total parasites (93.4% ±3.8%) (Fig 2D and 2E) and also had the highest infection intensity (Fig 2G). Skin parasites could be detected in any skin region, and in the chronic phase they were typically present in approximately 20 to 30 discrete foci of infection (Figs 2A and S1).

## Occurrence of a hindlimb spastic diplegia-like syndrome in the hamster infection model

Cardiomyopathy is the most serious adverse clinical consequence of chronic *T. cruzi* infection in humans. Histopathology analysis of heart tissue samples from hamsters at 5 months p.i. indicated that inflammation and collagen content (a measure of fibrosis) were not significantly increased compared to uninfected control animals at the group mean level (Fig 3). However, some animals did display clear evidence of oedema, focal and diffuse inflammatory foci, and intermuscular fibro-fatty tissue replacement.

To better understand the gait abnormality that arose in the hamsters, we assessed its development over time (Fig 4A). This identified excessive bilateral hindlimb adduction (movement towards the body's midline), which was clearly progressive over the course of chronic infection and resembled the "scissor gait" seen in a number of human neuromuscular disorders, such as cerebral palsy [38,39]. The phenotype was highly significant and developed in 100% of chronically infected hamsters (Fig 4D). Histopathological analysis of hindlimb skeletal muscle identified significant perivascular and muscle parenchymal inflammation, fibrosis and fatty degeneration (Fig 5). To investigate whether the gait abnormality might be linked to the myositis (muscle inflammation) and intense fibrosis, we investigated if these histopathological

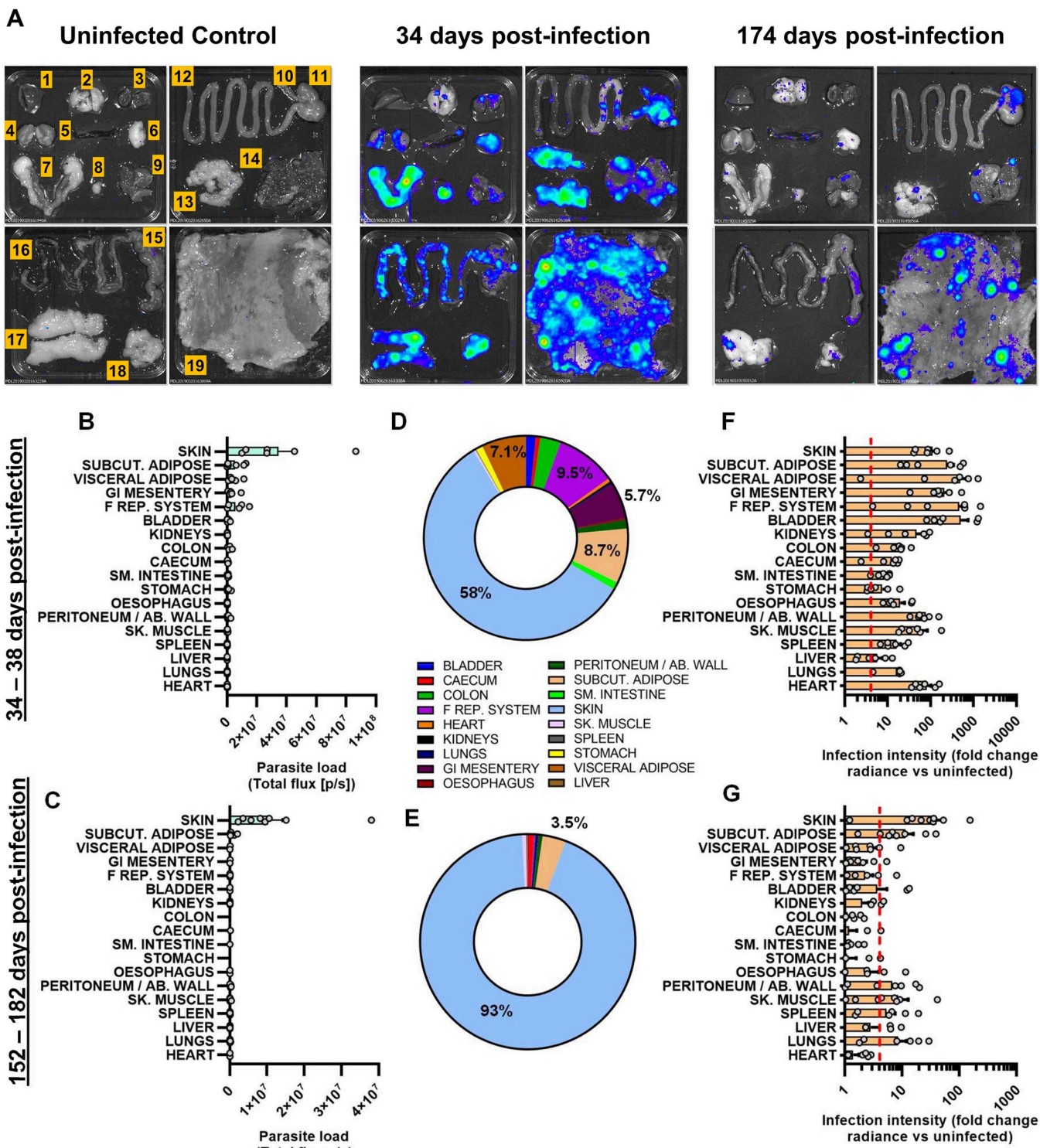

**Fig 2. Organ-specific infection profiles of *T. cruzi* CLBR in Syrian hamsters.** Quantification of organ/tissue-specific parasite loads and densities in TcVI-CLBR-infected Syrian hamsters using *ex vivo* bioluminescence imaging at 34–38 days (n = 6) and 152–182 days (n = 9) post-infection. Images (**A**) show bioluminescence intensities for one representative animal per group. Layout of samples shown on the uninfected control image: 1 = liver (partial), 2 = lungs, 3 = heart (bisected), 4 = kidneys, 5 = spleen, 6 = visceral adipose, 7 = female reproductive system, 8 = bladder, 9 = skeletal muscle (from hindleg), 10 = oesophagus, 11 = stomach, 12 = small intestine, 13 = gastro-intestinal mesentery, 14 = peritoneum/abdominal muscle, 15 = caecum, 16 = colon, 17 = subcutaneous adipose, 18 = visceral adipose, 19 = skin. **B,C**: Quantification of total bioluminescent flux (photons/sec) as a proxy of parasite loads for each organ and tissue type at 34–38 days (**B**) and 152–182 days (**C**) post-infection. **D,E**: Charts show the average proportional distributions of total measured *ex vivo*

parasite loads (total flux, p/s) across the analysed set of organ and tissue types at 34–38 days (**D**) and 152–182 days (**E**) post-infection. **F,G**: Quantification of bioluminescent radiance (photons/sec/cm$^2$/sr) as a proxy of infection intensity for each organ and tissue type expressed as fold change vs. matching organs or tissues from uninfected controls at 34–38 days (**F**) and 152–182 days (**G**) post-infection. Dashed line indicates detection threshold. In **B**, **C**, **F** and **G**, bars show the mean + S.E.M., and grey circles are data for individual hamsters.

features developed in a range of murine *T. cruzi* infection models (parasite strains TcI-JR and TcVI-CLBR in BALB/c, C57BL/6 and C3H/HeN mice), none of which develop the diplegia symptoms seen in hamsters. Significant myositis was present in 5 of the 6 parasite-mouse strain combinations, TcI-JR in C3H/HeN mice being the exception, and significant fibrosis was present in all 6 models (Fig 6). Both phenotypes occurred with a range of severities, with the most affected mouse models being broadly comparable to the hamster model: cellularity (myositis) increased 3-fold in TcVI-CLBR:C3H/HeN vs 4-fold for TcVI-CLBR:Hamster, and collagen content (fibrosis) increased ~4.5 fold in TcVI-CLBR:C3H/HeN vs 2-fold for TcVI-CLBR:Hamster. Fatty degeneration and parasite nests were also occasionally observed in murine skeletal muscle tissue. Thus, the diplegic gait in hamsters is highly unlikely to result

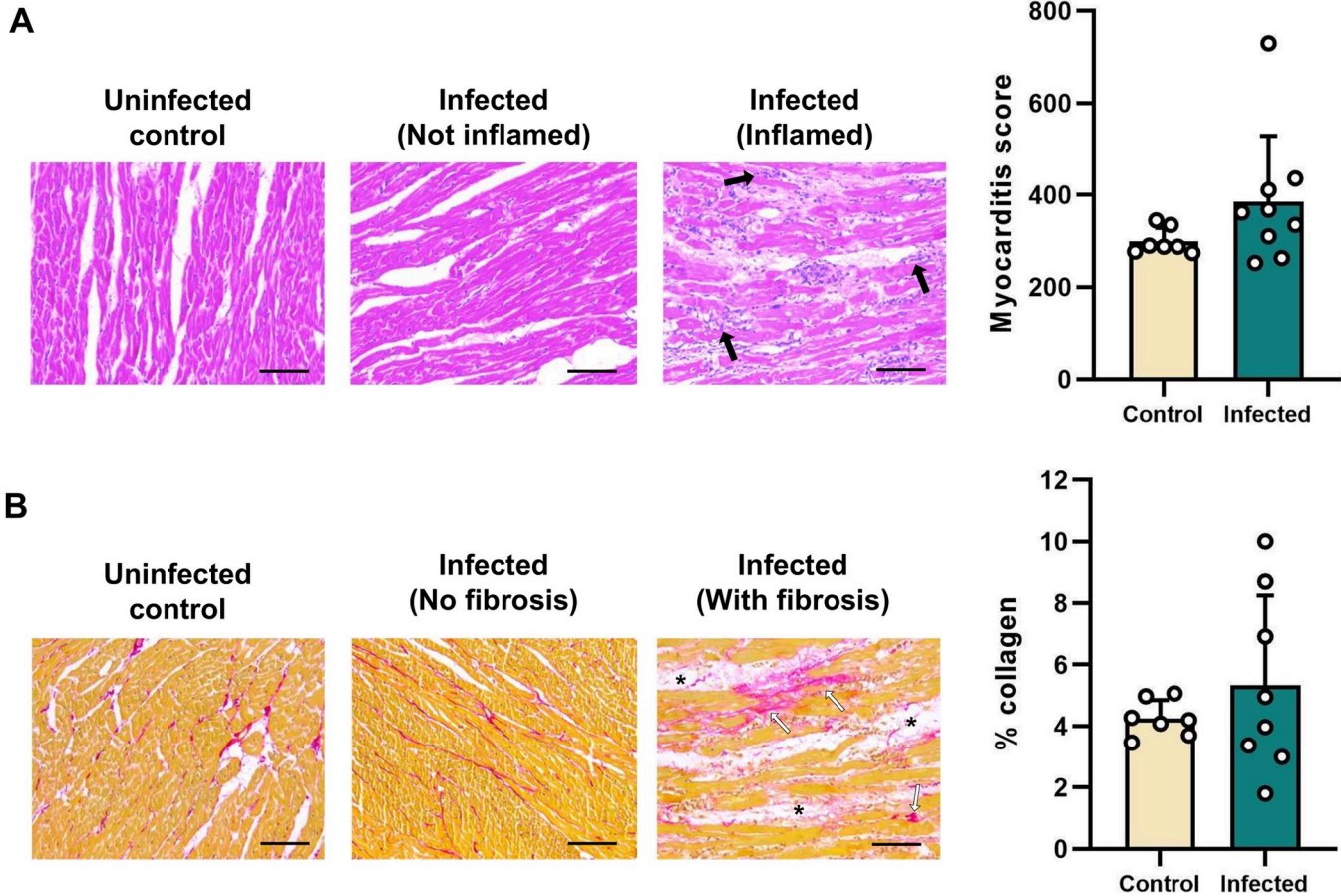

**Fig 3. Histopathological analysis of cardiac muscle tissue from *T. cruzi*-infected Syrian hamsters.** Quantification of cellularity (**A**) and collagen content (**B**) at 152–182 days post-infection as markers of chronic myocardial inflammation and fibrosis, respectively. Bars show the means + S.E.M., and grey circles are data for individual hamsters. Uninfected control (n = 7), infected (n = 9). Images show representative example sections stained with haematoxylin and eosin to reveal and inflammatory infiltrates (black arrows) (**A**), and sections stained with picro-sirius red to reveal excess collagen deposition (white arrows) and adipose replacement (asterisks) (**B**). Magnification 400X, scale bar = 50 μm.

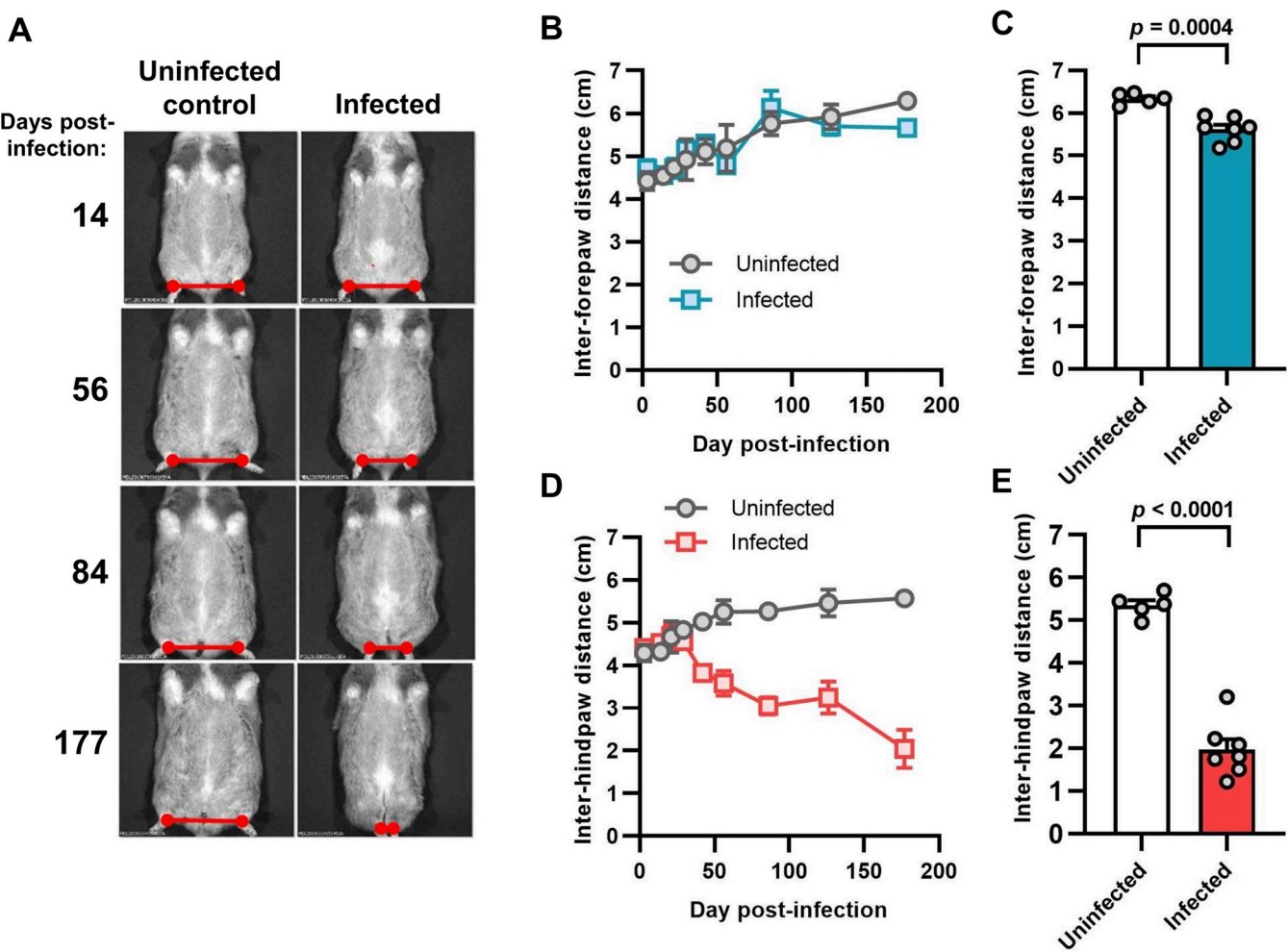

**Fig 4. Evaluation of limb adduction in *T. cruzi*-infected Syrian hamsters. A:** Example ventral view images of an individual hamster under transient anaesthesia (gaseous isoflurane) at the indicated day post-infection, with an age-matched uninfected control. Red marker lines highlight the inter-hindpaw distances. **B–E:** Quantification of the distance between forepaws (**B, C**) and hindpaws (**D, E**). In **B** and **D**, data are shown as the mean ± S.E.M. over time, uninfected control (n = 2), infected (n = 4). In **C** and **E**, data are the mean + S.E.M at the endpoint, with grey circles showing the measurements for individual animals, uninfected control (n = 5), infected (n = 7) (**C, E**).

directly from chronic parasite persistence, skeletal muscle inflammation or fibrosis, because these are all also common features of murine infections.

## Discussion

Bioluminescence imaging has been widely applied to *T. cruzi* murine infections, but its utility to monitor Chagas disease progression in larger rodents has not been reported. Our aims were to investigate if this technology could provide new insights into parasite distribution patterns during long-term *T. cruzi* infection of hamsters and to establish if there was a link with pathological outcomes. The results demonstrate that *in vivo* imaging, using the highly sensitive system described here, is able to facilitate longitudinal monitoring of chronic *T. cruzi* infections in hamsters. Furthermore, *ex vivo* imaging revealed organ and tissue parasite tropism during both acute and chronic stage infections, and identified the reservoir sites within the hamster that provide a niche for long-term parasite survival.

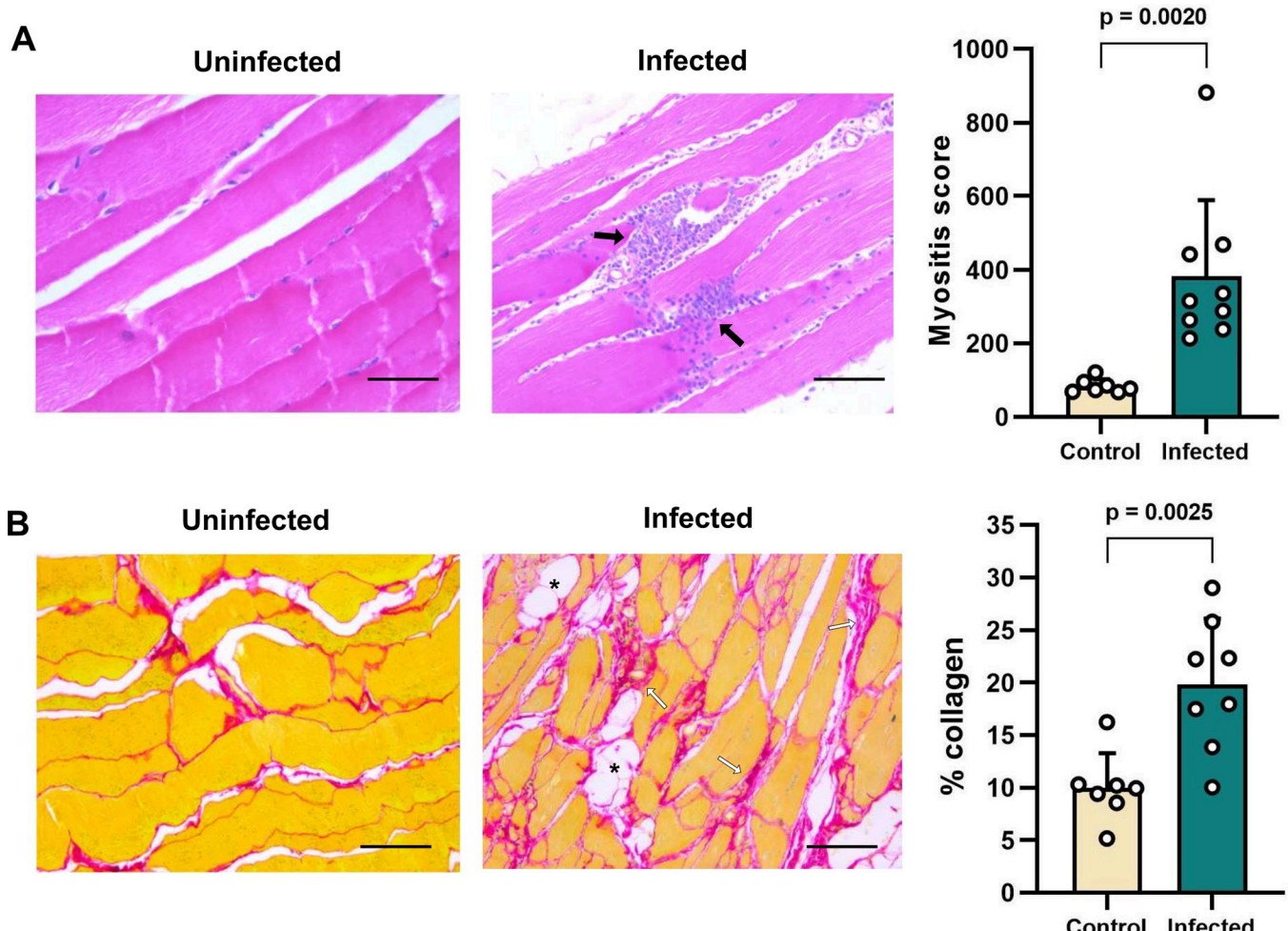

**Fig 5. Histopathological analysis of quadriceps tissue from *T. cruzi*-infected Syrian hamsters.** Quantification of cellularity (**A**) and collagen content (**B**) at 152–182 days post-infection as markers of chronic skeletal muscle inflammation and fibrosis, respectively. Bars show the means + S.E.M., and grey circles are data for individual hamsters. Uninfected control (n = 7), infected (n = 9). Images show representative example sections stained with haematoxylin and eosin to reveal and inflammatory infiltrates (black arrows) (**A**), and sections stained with picro-sirius red to reveal excess collagen deposition (white arrows) and adipose replacement (asterisks) (**B**). Magnification 400X, scale bar = 50 μm. *P*-values shown are from Student's *t*-tests.

There were both similarities and differences in comparison with typical infection profiles in mice, with one commonality being the pan-tropism of acute stage infections [34,40–44]. In terms of the heart, hamsters did not have detectable chronic infections localised to this organ and signs of cardiac pathology were only observed in a small subset of animals. Cardiac pathology typically is present in mice, although severity can vary substantially between parasite and mouse strains as, well as between individual animals, even when they are inbred [8–10,45]. The observation of a low rate of cardiomyopathy in hamsters may be significant, given that they are an outbred model, and only a minority of humans infected with *T. cruzi* develop heart disease symptoms [46]. Previous studies in hamsters have reported more consistent cardiomyopathy as well as mortality rates ranging from 12–80% [27–29,31–33,47]. However, here we observed 100% survival. This could reflect differences in virulence between strains because TcVI-CLBR also has low mortality, <10%, in mice in our laboratory [9]. In addition, the 5–6 month post-infection period to which our study was limited may have been too short for cardiac pathology to develop more reproducibly.

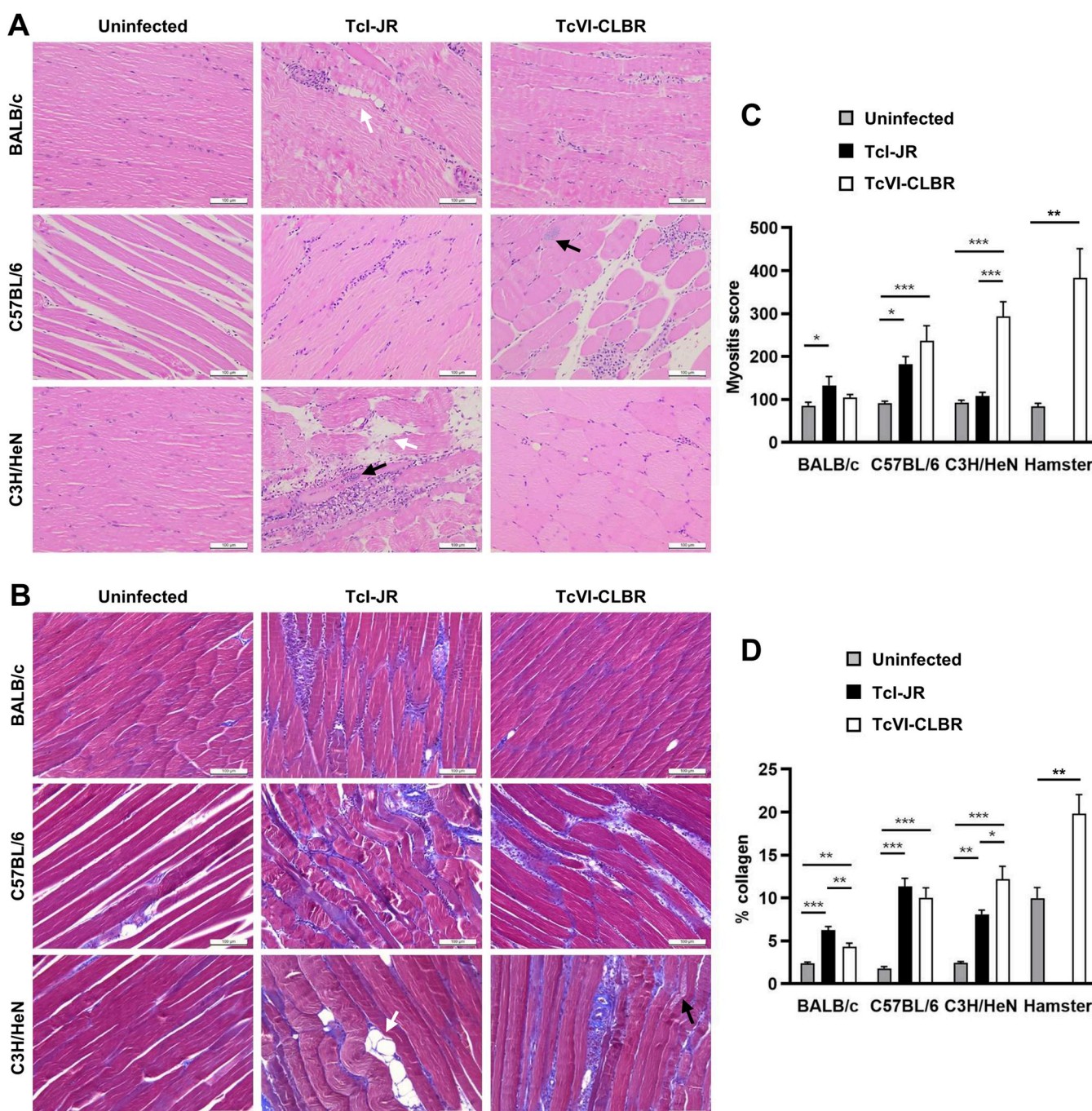

**Fig 6. Histopathological analysis of skeletal muscle tissue from multiple murine *T. cruzi* models. A:** Representative hindlimb muscle sections stained with haematoxylin and eosin, magnification 400X, scale bar = 100 μm. **B:** Quantitative histopathological analysis of hindlimb muscle samples obtained at 154–174 days post-infection from the following groups: TcVI-CLBR-BALB/c (n = 9), TcVI-CLBR-C57BL/6 (n = 10), TcVI-CLBR-C3H/HeN (n = 10), TcI-JR-BALB/c (n = 10), TcI-JR-C57BL/6 (n = 10), TcI-JR-C3H/HeN (n = 8), uninfected control BALB/c, C57BL.6 and C3H/HeN (all n = 10). Myositis score is the number of nuclei per image (6 x 10$^4$ μm$^2$). In **A** and **B**, black and white arrows indicate examples of parasite nests and areas of adipose replacement respectively. **C:** Representative hindlimb muscle sections stained with Masson's trichrome, magnification 400X, scale bar = 100 μm. **D:** Quantification of collagen content (% blue area) as a marker of skeletal muscle fibrosis severity in same groups as in **B**. Data are the means +S.E.M. and are from two independent experiments. Data for the equivalent analysis of tissue pathology in hamsters infected with TcVI-CLBR is shown for comparison. Asterisks indicate *p*-values for one way ANOVA comparisons between infection status groups, with each mouse strain analysed independently (* $P < 0.05$; ** $P < 0.01$; *** $P < 0.001$).

Regarding the digestive tract, localised chronic infections were absent, apart from a few very weak bioluminescence signals in one or two animals. No signs of organ enlargement or constipation, characteristic of digestive Chagas disease, were found. This contrasts with murine infections, where the GI tract is a major and almost universal site of long-term parasite persistence and chronic transit delays linked to the colon are significant in some parasite-mouse strain combinations [9,48–51]. GI tract samples from digestive Chagas disease patients have high frequencies of parasite DNA/antigen positivity [52–55]. Overall, these data indicate that Syrian hamster infection with TcVI-CLBR is not a suitable model to study the cardiac or the digestive form of Chagas disease.

Two findings from the study are particularly noteworthy. Firstly. the predominance of the skin as a chronic reservoir of infection, and secondly, the hindlimb diplegic gait phenotype. In mouse models, we have shown that chronic skin infection is detectable in 80–90% of mice across multiple combinations of *T. cruzi* and host genetic strains [19,37]. Whereas in mice the skin and GI tract have a similar reservoir status, in the hamster our data indicate that the skin and subcutaneous adipose tissue harbour almost the entire parasite burden in the animal. Acute cutaneous manifestations of Chagas disease reported in humans include localised inoculation 'chagomas'–swellings associated with initial parasite entry sites [2] and, very rarely, more disseminated lesions [56]. Skin lesions may also occur in cases of 're-activation' of a chronic infection in which immunosuppression, for example caused by HIV co-infection or post-transplant medications, causes expansion and spread of the parasite burden [57,58]. Although chronic *T. cruzi* infection in immunocompetent people is not associated with cutaneous pathology, this does not rule out the existence of a clinically silent, skin-resident parasite population. This would have an obvious evolutionary advantage for *T. cruzi* in terms of ensuring onward transmission to its blood-feeding insect vector and would be in keeping with the biology of related trypanosomatids, such as *T. brucei* and *Leishmania donovani* [59,60]. It may also be relevant that the skin is the most common site affected by adverse reactions to the antiparasitic drugs that are used to treat Chagas disease [61]. As well as establishing how frequent chronic skin infections are in humans, it will be important to investigate which cell types in the skin are parasitised and whether transmission to triatomines involves uptake of parasites from the tissue parenchyma or only the vasculature.

Skeletal muscle is a preferred site of infection for numerous human parasites e.g. *Sarcocystis*, *Toxoplasma*, *Brugia*, *Ancylostoma*, and *Trichinella* spp., [62]. Parasitism of this tissue is common in *T. cruzi* infected mice, though the frequency and intensity appears to be rather model-dependent [9,37,49,63]. We observed hindlimb skeletal muscle-localised parasite bioluminescence in all hamsters at 34–38 days p.i., and in five of nine (55.6%) hamsters at 152–182 days p.i., although the infection intensity was relatively low compared to other sites. This tissue was characterised by significant chronic parenchymal and perivascular inflammation and fibrosis, which was also present in all six of the mouse models that we analysed. The hindlimb hypertonia and diplegic gait were hamster-specific; no locomotion defects were observed in mice, implying that tissue parasitism, inflammation and fibrosis are insufficient explanations. A motor neuron-related aetiology is probably the most suitable hypothesis for future work. Others have reported limb paralysis in *T. cruzi*-infected mice [64,65], but that is clearly a different phenomenon from the one we have observed [66]. While neither type of locomotive dysfunction have been linked to *T. cruzi* infection in humans, there is evidence of peripheral nervous system involvement in some Chagas disease cohorts at frequencies of 10% [67] to 27% [68]. Skeletal muscle denervation, neuro-muscular junction, microvascular, mitochondrial and metabolic abnormalities have all been described [69–73], so it would be interesting to analyse whether these occur in *T. cruzi*-infected hamsters. Spastic diplegia in cerebral palsy is caused by congenital CNS lesions that impact on motor function [66] and *T. cruzi* infection

can extend to the CNS in severe cases, usually in association with immunosuppression [21,74]. A recent systematic review of literature encompassing 138 Chagas disease patients with CNS involvement found that 70% presented with a motor deficit, but only one case study specifically reporting spasticity [21].

In conclusion, further investigation of the mechanism underlying the gait defect we observe in our hamster model should consider both peripheral and central nervous system involvement. Neuropathological aspects of Chagas disease are evidently under-studied and the potential translational value of the hamster model developed in this study should be explored further. Lastly, this infection model may also prove relevant to wider research on spastic diplegia (bilateral lower limb muscle tightness), a common presentation of cerebral palsy that lacks a predictive small animal model [66].

## Supporting information

**S1 Fig. *Ex vivo* bioluminescence images for hindlimb skeletal muscle and skin samples from chronic *T. cruzi*-infected hamsters.** Images for each hamster were taken immediately post-mortem at at 152–182 days post-infection. Numbers are individual animal codes. Images from one representative not infected control are shown for comparison. Log-scale pseudocolour heat-map shows intensity of bioluminescence as a proxy for parasite load; minimum and maximum radiances are indicated. Red arrows highlight small foci of bioluminescence.
(TIF)

**S1 Data. Source files for 3D printed hamster anaesthesia delivery nosecone.**
(ZIP)

**S2 Data. Source data for figures in the manuscript.**
(XLSX)

## Acknowledgments

The authors thank the LSHTM Biological Services Facility and Veterinary staff for technical support and helpful discussions.

## Author Contributions

**Conceptualization:** Martin C. Taylor, John M. Kelly, Michael D. Lewis.

**Data curation:** Harry Langston, Amanda Fortes Francisco, Michael D. Lewis.

**Formal analysis:** Harry Langston, Amanda Fortes Francisco, Michael D. Lewis.

**Funding acquisition:** John M. Kelly, Michael D. Lewis.

**Investigation:** Harry Langston, Amanda Fortes Francisco, Ciaran Doidge, Archie A. Khan, Shiromani Jayawardhana, Michael D. Lewis.

**Methodology:** Harry Langston, Amanda Fortes Francisco, Michael D. Lewis.

**Project administration:** Martin C. Taylor, John M. Kelly, Michael D. Lewis.

**Resources:** Chrissy H. Roberts, Michael D. Lewis.

**Supervision:** John M. Kelly, Michael D. Lewis.

**Validation:** Harry Langston, Amanda Fortes Francisco, Ciaran Doidge.

**Visualization:** Harry Langston, Ciaran Doidge.

**Writing – original draft:** Michael D. Lewis.

**Writing – review & editing:** Harry Langston, Amanda Fortes Francisco, Archie A. Khan, Martin C. Taylor, John M. Kelly, Michael D. Lewis.

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
