## [Decision Letter · Decision Letter 0]

13 Jan 2024

Dear Dr Lewis,

Thank you very much for submitting your manuscript "Dynamics of *Trypanosoma cruzi* infection in hamsters and novel association with progressive motor dysfunction" for consideration at PLOS Neglected Tropical Diseases. As with all papers reviewed by the journal, your manuscript was reviewed by members of the editorial board and by several independent reviewers. In light of the reviews (below this email), we would like to invite the resubmission of a significantly-revised version that takes into account the reviewers' comments. 

The Reviewers and Editors acknowledge the relevance of your report showing the hamster as an alternative experimental model for Chagas disease. Even though cardiac and digestive track pathologies associated with human Chagas disease are not well reproduced in the hamster model, this new model offers new opportunities for studies of peripheral nervous system involvement in Chagas disease. This report is well within the scope of PNTD. However, please address all the comments and questions of the reviewers in your revised manuscript.

We cannot make any decision about publication until we have seen the revised manuscript and your response to the reviewers' comments. Your revised manuscript is also likely to be sent to reviewers for further evaluation.

Sincerely,

Alain Debrabant

Academic Editor

Charles Jaffe

Section Editor

The Reviewers and Editors acknowledge the relevance of your report showing the hamster as an alternative experimental model for Chagas disease. Even though cardiac and digestive track pathologies associated with human Chagas disease are not well reproduced in the hamster model, this new model offers new opportunities for studies of peripheral nervous system involvement in Chagas disease. This report is well within the scope of PNTD. However, please address all the comments and questions of the reviewers in your revised manuscript.

Reviewer's Responses to Questions

**Key Review Criteria Required for Acceptance?**

**Methods**

-Are the objectives of the study clearly articulated with a clear testable hypothesis stated?

-Is the study design appropriate to address the stated objectives?

-Is the population clearly described and appropriate for the hypothesis being tested?

-Is the sample size sufficient to ensure adequate power to address the hypothesis being tested?

-Were correct statistical analysis used to support conclusions?

-Are there concerns about ethical or regulatory requirements being met?

Reviewer #1: -Are the objectives of the study clearly articulated with a clear testable hypothesis stated?

YES

-Is the study design appropriate to address the stated objectives?

YES

-Is the population clearly described and appropriate for the hypothesis being tested?

YES

-Is the sample size sufficient to ensure adequate power to address the hypothesis being tested?

YES

-Were correct statistical analysis used to support conclusions?

YES

-Are there concerns about ethical or regulatory requirements being met?

NO

Reviewer #2: -Are the objectives of the study clearly articulated with a clear testable hypothesis stated? yes

-Is the study design appropriate to address the stated objectives? yes

-Is the population clearly described and appropriate for the hypothesis being tested? yes

-Is the sample size sufficient to ensure adequate power to address the hypothesis being tested? yes

-Were correct statistical analysis used to support conclusions? yes

-Are there concerns about ethical or regulatory requirements being met? No

For the fibrosis index information is missing as to how many fields were analysed for each image. For both the inflammatory and fibrosis indices it is not clear if all fields of view were in the same tissue slice or if multiple tissue slices were analysed per animal. Please add this information in the methods (lines 142 – 148).

Line 144: “inflammation index” – I assume this is the same as the myositis score, please clarify/make consistent in the methods section.

**Results**

-Does the analysis presented match the analysis plan?

-Are the results clearly and completely presented?

-Are the figures (Tables, Images) of sufficient quality for clarity?

Reviewer #1: -Does the analysis presented match the analysis plan?

YES

-Are the results clearly and completely presented?

YES

-Are the figures (Tables, Images) of sufficient quality for clarity?

YES

Reviewer #2: -Does the analysis presented match the analysis plan? yes

-Are the results clearly and completely presented?

Line 206: limb adduction: I think it would benefit the reader to give a brief explanation here of what limb adduction is.

Line 277: hindlimb parasite bioluminescence in half of the animals at day 152-182. This data does not appear to be presented in the results section. Considering the importance of this data with respect to the adduction phenotype, hindlimb skeletal muscle associated bioluminescence should be shown in Figure 2 B to G, or as a separate panel in figure 4.

-Are the figures (Tables, Images) of sufficient quality for clarity? Yes

**Conclusions**

-Are the conclusions supported by the data presented?

-Are the limitations of analysis clearly described?

-Do the authors discuss how these data can be helpful to advance our understanding of the topic under study?

-Is public health relevance addressed?

Reviewer #1: -Are the conclusions supported by the data presented?

YES

-Are the limitations of analysis clearly described?

YES

-Do the authors discuss how these data can be helpful to advance our understanding of the topic under study?

PARTIALLY

-Is public health relevance addressed?

PARTIALLY

Reviewer #2: -Are the conclusions supported by the data presented? Yes, except: the authors show inflammation and fibrosis in the quadriceps of infected hamsters. To determine if inflammation and fibrosis are directly linked to the gait abnormalities in infected hamsters the authors checked mouse muscle tissues for inflammation and fibrosis, and found clear markers of inflammation and fibrosis in mouse myocardium. As mice do not show gait abnormalities, that authors conclude that the gait abnormalities are likely not the direct result of parasite induced inflammation and fibrosis. However, different tissues are being assessed in mice compared to hamsters. The relevant tissue for the gait abnormality is the skeletal muscle of the hindlimb (e.g. quadriceps). For the argument to be robust the authors should investigate the levels of inflammation and fibrosis in mouse quadriceps, as it is possible that infected mice do not show inflammation and fibrosis in their hindlimbs. I suggest that either histopathological data for mice quadriceps are included, or a caveat is included in the discussion.

-Are the limitations of analysis clearly described? A limitation of the study is that only one strain was assessed. The value of carrying additional studies with different strains should be covered briefly in the discussion. 

-Do the authors discuss how these data can be helpful to advance our understanding of the topic under study? yes

-Is public health relevance addressed? Yes

**Editorial and Data Presentation Modifications?**

Reviewer #1: (No Response)

Reviewer #2: According to the PLOS data availability policy the following data should be provided:

• The values behind the means, standard deviations and other measures reported;

• The values used to build graphs;

This data is not provided for the manuscript.

**Summary and General Comments**

Reviewer #1: SUMMARY

The aim of this study was to develop an in vivo/ex vivo bioluminescence imaging system for Trypanosoma cruzi (T. cruzi) infection in a hamster model.

The mouse animal model has been widely used in Chagas disease (CD) research. Chronically T. cruzi infected mice develop cardiac and gastrointestinal pathologies reproducing some clinical features of the disease in humans. 

A better understanding of Chagas disease pathogenesis was possible with the development of highly sensitive bioluminescence imaging based on the use of T. cruzi parasites that express a red-shifted luciferase. These investigations allowed the study of parasite dissemination and infection dynamics in mice. 

In this paper, the authors applied the bioluminescence imaging technology to study T. cruzi infection in the hamster model, which shows a close alignment with human diseases for several pathogens such as Leishmania donovani among others. 

In contrast to what has been found in the mouse model, chronically infected hamsters did not show signs of heart or gut disease. The infection was restricted to the skin and subcutaneous adipose tissue. Interestingly, they exhibited stiffness in the hindlimbs that resulted in an altered gait, suggestive of nervous system damage. 

MERITS

Few research groups have studied CD in the hamster model. This is the first investigation that monitors T. cruzi infection over time and clinical manifestations in large rodents, using bioluminescence imaging. Although preliminary, this work provides evidence of peripheral nervous system damage in chronic CD and may be valuable for future research on this unique and not well studied pathological outcome of the disease. However, the translational value of this model needs to be further explored, mainly because humans do not develop similar locomotive dysfunctions. 

COMMENTS

1. Introduction:

a) Line 64: please include a few references about the models suitable for studying aspects of Chagas digestive disease.

b) Line 69: as the authors mentioned, one manifestation of CD is the presence of parasites in the central nervous system in the acute phase or due to reactivation during chronic disease. Indeed, patients with cerebral CD may have symptoms such as confusion, headaches, convulsions and altered mental status (recently reviewed by Shelton WJ, Gonzalez JM. “Outcomes of patients in Chagas disease of the central nervous system: a systematic review”. Parasitology. 2023 Nov 21:1-9. doi: 10.1017/S0031182023001117). I suggest the authors to expand this paragraph and add information on this important clinical aspect involving the nervous system in humans.

2. Methods: 

c) Line 109: please indicate the source of the Living Image v4.7 software.

d) Lines 121-122: please provide a link or reference for the detection threshold method.

e) Line123: please indicate which version of the Living Image software you used.

f) Line 144: please provide a link or reference for the inflammation index calculation.

g) Line 146: please provide a link or reference for the fibrosis index calculation.

h) Please add a sub-section to describe how you evaluated the limb abduction with references.

3. Results: 

i) Figure 3: please indicate/circle the inflammation and fibrosis regions in the infected heart tissue.

j) Line 208: the authors state that “the phenotype was highly significant and developed in 100% of chronically infected hamsters”. The results shown in Figure 4B do not reflect that statement. Please explain. 

k) Line 223: please clarify and expand the sentence “A neuropathy aetiology therefore seems probable”.

l) Figure 5: same as in i).

m) Figure 6: please add the bars corresponding to the myositis score and % collagen obtained in the hamster study to facilitate the comparison.

4. Discussion:

n) Line 269: the authors acknowledge the evolutionary advantage of skin-resident parasites populations and the impact on T. cruzi transmission. They have found that the skin is a predominant parasite reservoir in chronically infected hamsters. This finding is important in the context of CD transmission and highlights the utility of the hamster model. I encourage the authors to expand this paragraph to mention this remarkable observation. 

o) Line 290: what is spastic displegia? why will research on spastic displegia be relevant? Please explain and cite a reference. 

p) Please expand the last paragraph to further explain why you think this model could be useful to explore peripheral nervous system damage in chronic CD although humans do not develop similar symptoms.

Reviewer #2: In this manuscript the authors develop a bioluminescent hamster model to assess its value for Chagas disease research, in particular with respect to parasite distribution and pathological outcomes. The study shows that longitudinal imaging of the infection is possible in hamsters, although detection limit may be worse than in mice due to larger size of the animals and organs. In the acute phase parasites are widely distributed as seen in other models. Interestingly in the chronic phase parasites reside mainly in skin, and also other fatty tissues, and the GI tract does not appear to be a reservoir unlike in mice.

In line with the distribution pattern, a low rate of cardiomyopathy was detected, lower than seen in mice. While this may align with the human situation where cardiomyopathy is only detected in a subset of people, a question is how much this a Tc CLBR strain specific effect. As mentioned by the authors previous studies with hamsters have shown more frequent cardiomyopathy. 

Overall, the presented model is not deemed suitable for the study of cardiac and digestive Chagas pathology. However, the study did identify a previously not observed Chagas disease pathology affecting the hindlimbs and gait. While this is not a symptom of human disease there may be value in understanding the underlying pathology and its translational value.

This work is of interest to the general Chagas disease community, including those interested in models of human pathophysiology and drug discovery.

PLOS authors have the option to publish the peer review history of their article (<a href="https://journals.plos.org/plosntds/s/editorial-and-peer-review-pro

---

## [Decision Letter · Decision Letter 1]

7 Jun 2024

Dear Dr Lewis,

We are pleased to inform you that your manuscript 'Dynamics of *Trypanosoma cruzi* infection in hamsters and novel association with progressive motor dysfunction' has been provisionally accepted for publication in PLOS Neglected Tropical Diseases.

Best regards,

Alain Debrabant

Academic Editor

Hira Nakhasi

Section Editor

<style type="text/css">p.p1 {margin: 0.0px 0.0px 0.0px 0.0px; line-height: 16.0px; font: 14.0px Arial; color: #323333; -webkit-text-stroke: #323333}span.s1 {font-kerning: none

</style>

Reviewer's Responses to Questions

**Key Review Criteria Required for Acceptance?**

**Methods**

-Are the objectives of the study clearly articulated with a clear testable hypothesis stated?

-Is the study design appropriate to address the stated objectives?

-Is the population clearly described and appropriate for the hypothesis being tested?

-Is the sample size sufficient to ensure adequate power to address the hypothesis being tested?

-Were correct statistical analysis used to support conclusions?

-Are there concerns about ethical or regulatory requirements being met?

Reviewer #1: -Are the objectives of the study clearly articulated with a clear testable hypothesis stated? YES

-Is the study design appropriate to address the stated objectives? YES

-Is the population clearly described and appropriate for the hypothesis being tested? YES

-Is the sample size sufficient to ensure adequate power to address the hypothesis being tested? YES

-Were correct statistical analysis used to support conclusions? YES

-Are there concerns about ethical or regulatory requirements being met? YES

Reviewer #2: (No Response)

**Results**

-Does the analysis presented match the analysis plan?

-Are the results clearly and completely presented?

-Are the figures (Tables, Images) of sufficient quality for clarity?

Reviewer #1: -Does the analysis presented match the analysis plan? YES

-Are the results clearly and completely presented? YES

-Are the figures (Tables, Images) of sufficient quality for clarity? YES

Reviewer #2: (No Response)

**Conclusions**

-Are the conclusions supported by the data presented?

-Are the limitations of analysis clearly described?

-Do the authors discuss how these data can be helpful to advance our understanding of the topic under study?

-Is public health relevance addressed?

Reviewer #1: -Are the conclusions supported by the data presented? YES

-Are the limitations of analysis clearly described? YES

-Do the authors discuss how these data can be helpful to advance our understanding of the topic under study? YES

-Is public health relevance addressed? YES

Reviewer #2: (No Response)

**Editorial and Data Presentation Modifications?**

Reviewer #1: (No Response)

Reviewer #2: (No Response)

**Summary and General Comments**

Reviewer #1: The authors have provided a detailed and thorough response to my comments/questions. They have adequately addressed my concerns. I have no further questions and I believe the manuscript can be accepted for publication.

Reviewer #2: (No Response)

PLOS authors have the option to publish the peer review history of their article (what does this mean?). If published, this will include your full peer review and any attached files.

Reviewer #1: No

Reviewer #2: No

---

## [Editor Report · Acceptance letter]

15 Jun 2024

Dear Dr Lewis,

We are delighted to inform you that your manuscript, "Dynamics of *Trypanosoma cruzi* infection in hamsters and novel association with progressive motor dysfunction," has been formally accepted for publication in PLOS Neglected Tropical Diseases.

Best regards,

Shaden Kamhawi

co-Editor-in-Chief

Paul Brindley

co-Editor-in-Chief
